# A Closer Look at the Dermatological Profile of GLP-1 Agonists

**DOI:** 10.3390/diseases13050127

**Published:** 2025-04-22

**Authors:** Calista Persson, Allison Eaton, Harvey N. Mayrovitz

**Affiliations:** 1Dr. Kiran Patel College of Osteopathic Medicine, Nova Southeastern University, Davie, FL 33328, USA; cp2109@mynsu.nova.edu (C.P.); ae1100@mynsu.nova.edu (A.E.); 2Dr. Kiran Patel College of Allopathic Medicine, Nova Southeastern University, Davie, FL 33314, USA

**Keywords:** GLP-1 receptor agonists, dermatological side effects, inflammatory skin diseases, hypersensitivity reactions, psoriasis, hidradenitis suppurativa, injection site reactions, immune modulation, metabolic changes, weight loss

## Abstract

Background/objectives: Glucagon-like peptide-1 receptor agonists (GLP-1RAs) are widely used in treating type 2 diabetes and obesity, offering established metabolic and cardiovascular benefits. Emerging evidence suggests these agents also exert direct dermatologic effects. This systematic review categorizes these effects and explores their role in inflammatory skin diseases. Methods: A comprehensive literature search was performed across EMBASE, PubMed, Web of Science, and Google Scholar for studies published from 2014 to 2025. Inclusion criteria were English-language, peer-reviewed original research involving human subjects that linked GLP-1RAs to dermatologic effects. Animal and in vitro studies were excluded. PRISMA guidelines were followed. Results: Fifty-one studies met inclusion criteria. Thirty-four reported adverse effects, including hypersensitivity, injection-site reactions, pruritus, urticaria, angioedema, and immune-mediated conditions like bullous pemphigoid. Seventeen studies described beneficial outcomes, such as improvements in psoriasis, reduced hidradenitis suppurativa flares, enhanced wound healing, anti-aging potential, and decreased inflammation. GLP-1RAs showed cytokine modulation in psoriasis, though their role in hidradenitis suppurativa remains uncertain. Cosmetic concerns, such as “Ozempic Face” due to rapid weight loss, were also noted. Conclusions: GLP-1RAs have a broad spectrum of dermatologic effects, from immunomodulatory benefits to adverse cutaneous reactions. Their impact on inflammatory skin disorders suggests a novel therapeutic avenue. However, adverse reactions and aesthetic changes warrant vigilance. Future research should focus on mechanistic studies, long-term safety, and identifying biomarkers to predict dermatologic responses, ultimately guiding personalized treatment approaches.

## 1. Introduction

Glucagon-like peptide-1 receptor agonists (GLP-1RAs) function as medications that replicate the activity of endogenous GLP-1, which intestinal L-cells produce as an incretin hormone after nutrient consumption [1]. GLP-1 plays a central role in managing glucose levels while also regulating appetite and supporting cardiovascular health [2]. These medications activate GLP-1 receptors (GLP-1Rs) located in the pancreas and the central nervous system as well as in the gastrointestinal tract to trigger insulin secretion enhancement while reducing glucagon release and slowing down gastric emptying, which results in better glycemic control and substantial weight loss [3].

GLP-1RAs function in pancreatic beta cells to stimulate insulin secretion that depends on glucose levels, thus releasing insulin only when glucose concentrations rise and minimizing hypoglycemia risks [4]. These medications stop glucagon from being released by alpha cells, which helps prevent the liver from producing too much glucose. Dual actions from GLP-1RAs result in decreased fasting and postprandial blood glucose levels, which are fundamental for diabetes management [5].

GLP-1Rs located in the hypothalamus suppress appetite by activating satiety pathways, in addition to their role in glycemic control. The suppression of food intake and resulting weight loss provides essential advantages for patients who struggle with obesity or complications from excessive weight [6]. GLP-1RAs delay gastric emptying, which slows nutrient absorption and stabilizes glucose levels after meals [7].

GLP-1RAs serve multiple medical purposes. They treat type 2 diabetes mellitus and help manage weight while reducing cardiovascular risks. Major clinical trials demonstrate that these drugs lower major adverse cardiovascular events like myocardial infarction and stroke [8].

The medical use of GLP-1RAs started with T2DM treatment but now includes obesity management for patients whose BMI is ≥30 kg/m^2^ or ≥27 kg/m^2^ who also have weight-related health problems [9]. Semaglutide and liraglutide usage in weight control produces sustained and meaningful weight reduction, which allows these drugs to become potential surgical alternatives for bariatric procedures in select patient groups [10,11].

GLP-1RAs all replicate native GLP-1 yet exist as individual modified peptides with distinct structural properties as single molecular entities instead of drug combinations. Exenatide represents an artificial 39-amino-acid version of exendin-4 derived from Gila monster venom, which contains roughly 50% sequence similarity to human GLP-1. Liraglutide and semaglutide represent human GLP-1(7–37) analogs, which have specific amino acid changes and fatty-acid side chains to extend their duration of action. Liraglutide retains ~97% homology to GLP-1 with a Lys→Arg substitution at position 34 while featuring a C16 palmitic acid attached to Lys^26^ enabling albumin binding [12]. Semaglutide shares ~94% homology with GLP-1 through an Ala^8^→2-aminoisobutyric acid (Aib8) substitution, which protects against DPP-4 degradation and includes a C18 diacid fatty acyl chain at Lys^26^, which both enhances albumin binding and extends its half-life. Dulaglutide represents a larger protein conjugate: The structure consists of two GLP-1 analog sequences covalently connected through a short peptide to an IgG4 Fc fragment, creating a dimeric fusion protein that demonstrates resistance to clearance. The molecular modifications result in different pharmacokinetic profiles between short-acting medications like exenatide BID and long-acting weekly treatments such as dulaglutide and semaglutide. However, they activate receptors through similar mechanisms [12].

GLP-1 receptor agonists attach to GLP-1 receptors on pancreatic β-cells, triggering adenylate cyclase activation and elevating cAMP levels inside the cells. The process boosts glucose-dependent insulin production while reducing glucagon release from the pancreatic α-cells, resulting in decreased blood sugar levels and a minimized risk of hypoglycemia because of increased insulin and decreased glucagon secretion [13]. Apart from pancreatic effects, GLP-1Rs are located in the brainstem and hypothalamus, where their activation leads to increased satiety and reduced appetite. GLP-1RAs inhibit gastric emptying by acting on receptors located in the gastrointestinal tract. These combined mechanisms help better manage blood sugar while aiding weight loss (Figure 1) [13].

GLP-1RAs used in clinical settings are provided as peptide solutions with high purity supported by precisely selected excipients to ensure their stability and bioactivity. The multi-dose pens for liraglutide and semaglutide contain buffering agents such as disodium phosphate to maintain a neutral pH. The formulation includes disodium phosphate as an isotonicity agent with propylene glycol and phenol as a preservative to inhibit microbial growth [11].

Exenatide uses an acetate buffer mixed with mannitol for tonicity maintenance, while m-cresol acts as its preservative [11]. The once-weekly auto-injectable dulaglutide comes in a single-use prefilled syringe, which contains no preservatives but includes excipients like citrate buffer, mannitol, and polysorbate-80 to maintain the large fusion protein’s stability and prevent aggregation. Although these excipients provide no therapeutic benefit, they help preserve the peptide’s structural integrity and activity throughout storage. The bioactivity of GLP-1RA remains preserved throughout the product’s shelf-life and usage period due to the high purity of the active ingredient (>98–99%) and antimicrobial agents phenol or m-cresol in multi-dose vials [11].

GLP-1RAs’ cardiovascular advantages have resulted in their preference as second-line treatment for T2DM patients who already have established atherosclerotic cardiovascular disease. Recent research indicates that GLP-1RAs may protect kidneys by decreasing albuminuria and maintaining renal function, which shows their potential usefulness in treating diabetic kidney disease [14].

GLP-1RAs offer therapeutic benefits but also cause multiple adverse effects. Gastrointestinal side effects like nausea, vomiting, diarrhea, and constipation represent the most common adverse reactions and typically happen during dose escalation [15]. Although infrequent, severe complications such as pancreatitis and gallbladder disease along with thyroid C-cell hyperplasia have been observed, and rodent research indicates a potential risk for developing medullary thyroid carcinoma [10].

GLP-1RAs produce more frequent and severe side effects when administered through larger doses or quicker titration schedules. The frequency of nausea displayed by patients shows dose dependency because the occurrence approximately doubles when higher doses of the same medication are administered compared to lower doses. In contrast, patients generally report only mild-to-moderate nausea [15]. Exenatide 10 µg BID increased nausea incidence significantly more than exenatide five µg BID in clinical trials, as indicated by an OR of approximately 2.3 for nausea. The rate of nausea/vomiting was higher for patients taking liraglutide 1.8 mg compared to those taking 1.2 mg. Therapy continuation leads to a reduction in effects because tolerance development occurs over time. Low-dose GLP-1RAs are initiated at the start of treatment and gradually increased to enhance tolerability. The administration of liraglutide starts with a 0.6 mg daily dose for one week to minimize initial GI symptoms before moving up to doses of 1.2 mg or higher [15]. Gradually increasing dosage helps avoid sudden adverse effects from drug doses [16]. Proper dosing schedules become essential because higher doses lead to more frequent side effects such as gastrointestinal upset, and the side effects can be managed by using the minimum effective dose or slow titration (Figure 2).

The dermatological effects caused by GLP-1RAs have recently become a significant topic for medical professionals. Patients treated with these medications can develop several skin reactions, including injection-site reactions, hypersensitivity responses, pruritus, rashes, and immune-mediated skin disorders such as psoriasis (Pso) and hidradenitis suppurativa (HS) [14,17,18,19,20]. Most dermatological reactions to these treatments resolve without intervention, but some patients experience chronic skin complications that need medical attention [14,17].

Recent findings indicate that GLP-1RAs affect inflammatory skin diseases beyond their immediate adverse effects. Research shows that these medications can regulate inflammatory processes by decreasing pro-inflammatory cytokines such as tumor necrosis factor-alpha (TNF-α) and interleukin-17 (IL-17) [8,19]. Connections between metabolic disorders and skin diseases such as Pso and HS lead to varying effects of GLP-1RAs based on individual immune reactions [18].

GLP-1RA therapy leads to weight reduction and produces effects both in terms of aesthetics and dermatology. The “Ozempic Face” term refers to the significant facial volume loss that occurs when rapid fat reduction depletes subcutaneous fat [21]. The cosmetic effect of rapid fat reduction has led to increased interest among patients in corrective dermatological treatments, including dermal fillers and skin-tightening procedures [11,22].

The effects of GLP-1RAs on hair and nail health represent an emerging field of research. Case studies indicate GLP-1RAs could cause hair thinning or shedding potentially due to rapid weight loss and nutritional deficiencies or because of direct effects on hair follicles [23]. Some patients have shown altered nail growth and increased nail fragility, but extensive research is required to confirm this link [24,25].

Research has been conducted into how GLP-1RAs affect skin aging processes and their capability to heal wounds. Scientists suspect GLP-1-related drugs may affect collagen production and skin regeneration because they are involved in cellular repair mechanisms [26]. Initial research indicates GLP-1RAs may improve wound healing through increased angiogenesis and fibroblast activity, especially in diabetic patients with tissue repair difficulties [27]. According to research findings, further clinical studies are necessary to validate if these therapeutic effects of GLP-1RAs apply to broader dermatological uses, including anti-aging treatments [28].

New research shows possible links between GLP-1RAs and a wider range of autoimmune skin diseases besides Pso and HS. This implies that GLP-1RAs may contribute to the treatment of immune dysregulation disorders, including lupus erythematosus and dermatomyositis [2]. Studies have recently indicated that GLP-1RAs might influence melanogenesis, which may impact hyperpigmentation disorders and vitiligo. Preliminary data show that GLP-1 signaling affects melanin creation in both keratinocytes and melanocytes, which presents a new direction for future research despite limited available information [3]. The long-term effects of drugs that target systemic inflammation and metabolic pathways on autoimmune skin diseases needs further investigation.

GLP-1RAs represent an effective treatment option for type 2 diabetes and obesity patients and demonstrate established cardiovascular and nephroprotective advantages [13]. Clinical studies continue to increase, yet the dermatological effects of these treatments remain poorly understood and lack adequate characterization. This study seeks to systematically assess and combine existing human-based findings about skin conditions linked to GLP-1RA therapy. This review fills an essential gap in the literature by examining and categorizing skin-related adverse and beneficial outcomes through frequency and severity assessments. This investigation examines how GLP-1RAs may connect to inflammatory skin conditions through immunometabolic pathways. This publication presents the first extensive analysis that unifies the dermatologic effects of various GLP-1RA therapies across different treatment scenarios while delivering new knowledge that benefits interdisciplinary patient care.

## 2. Materials and Methods

The systematic review encompasses 51 studies comprising 22 case studies, 2 clinical trials, 2 cohort studies, 10 systematic reviews and meta-analyses, 2 scoping reviews, and 13 narrative reviews. The research examines how GLP-1RAs like exenatide, liraglutide, dulaglutide, and semaglutide affect skin health. The study investigates patients with T2DM and obesity who have immune-mediated dermatologic conditions using randomized controlled trials, cohort studies, case reports, and systematic reviews. The research provides an extensive summary of the skin effects associated with GLP-1RA treatment, including adverse reactions and potential medical advantages.

### 2.1. Literature Search Strategy

Researchers performed a complete literature search through EMBASE, PubMed, Web of Science, and Google Scholar to find studies published from 2014 to 2025. These databases were selected for their complementary strengths: PubMed specializes in biomedical research. At the same time, EMBASE covers a broader range of pharmacological studies and European literature as Web of Science provides extensive multidisciplinary indexing with citation tracking capabilities, and Google Scholar captures grey literature and non-indexed sources. This combination enabled comprehensive and equitable retrieval of peer-reviewed research articles. The chosen period captures the newest GLP-1RA research developments while omitting outdated studies with little clinical relevance today.

The research search incorporated terms such as “GLP-1 agonists”, “dermatological side effects”, “rash”, “skin reactions”, “pruritus”, “exenatide”, “liraglutide”, “semaglutide”, and “side effects”. The research team applied Boolean operators (AND, OR) to filter search outcomes while they checked reference lists manually to find more studies [29].

Across four databases, the initial search produced 208 citations. The initial screening process eliminated 157 articles according to established inclusion and exclusion criteria, resulting in 51 studies being selected for detailed review and final analysis. Given the evolving nature of this novel topic, this systematic review included well-researched systematic reviews and meta-analyses—sourced both internationally and from less readily accessible databases—as they contributed a greater depth and breadth of information. The PRISMA diagram (see Figure 3) served as a visual representation to track and clarify study selection while guaranteeing transparency, reproducibility, and reducing bias for this systematic review [30].

### 2.2. Inclusion and Exclusion Criteria

The inclusion criteria were as follows:
The research encompassed original studies and clinical trials, alongside observational studies, case reports, case series, meta-analyses, and recent comprehensive systematic reviews, all examining the dermatological effects of GLP-1RAs in humans aged 18 and older.Research articles that document negative skin responses or possible medicinal benefits of GLP-1RAs.Publications available in English.Peer-reviewed studies

The exclusion criteria were as follows:
The review excluded opinion pieces.Studies involving animal subjects.The review excluded studies that failed to investigate GLP-1RAs’ dermatological impact.

### 2.3. Data Extraction and Synthesis

Researchers used a systematic yet personalized method to extract data from each study. Two independent reviewers conducted a systematic analysis of the selected studies and compiled summaries of their objectives, methodologies, sample characteristics, and key findings and conclusions. We resolved differences between reviewers through discussion and sought input from a third reviewer when required.

We conducted hand searches and reviewed reference lists from relevant articles for potentially relevant studies. A qualitative synthesis of findings revealed dermatological patterns and trends related to various GLP-1RA treatments. Research teams performed sub-group analyses to assess how side effects changed according to the GLP-1RA medication administered.

### 2.4. Quality Assessment

Critical analysis was used to evaluate included studies, which verified their credibility, reproducibility, and unbiased nature. The evaluation process included examining study design details along with sample size metrics and data collection techniques while also assessing sources of bias and methods of statistical analysis and reproducibility. This review method maintains that only verified high-quality evidence informs our knowledge about the effects of GLP-1RAs on dermatology.

The synthesis of findings is presented in the following tables: The report includes three key tables that capture comprehensive information on GLP-1RAs’ skin-related impacts: (1) outlines adverse dermatological effects along with their reported incidence rates; (2) compares skin side effects among various GLP-1RAs to identify differences between medications; and (3) summarizes dermatological benefits including research references supporting these effects. The following analysis in the Results section depends on these tables, which explore specific patterns, trends, and implications of dermatological effects related to GLP-1RAs.

## 3. Results

GLP-1RAs have revealed dermatological associations, which have gained research interest because multiple reports indicate adverse skin reactions. The provided data represent only documented cases from specific research studies, which might not reflect the true prevalence rates of these events. Differences in study design and reporting practices affect these numbers.

Our research uncovered no large-scale clinical studies regarding the dermatological effects of GLP-1RA use. Since no large-scale clinical data were available, our analysis depended mainly on single case studies and restricted clinical information.

A total of 51 studies fulfilled the inclusion criteria. Among these 51 studies, 34 demonstrated a direct connection between GLP-1RAs and adverse dermatological effects, which included hypersensitivity reactions and injection-site reactions together with pruritus urticaria and angioedema and some rare immune-mediated conditions like bullous pemphigoid (BP). The analysis demonstrated that 17 studies found GLP-1RAs produced advantageous dermatological effects, including psoriasis improvement alongside reduced HS flare-ups, improved wound healing potential, and anti-aging benefits while reducing skin inflammation.

The analysis demonstrates the necessity for extensive clinical research to fully explore the dermatological effects of GLP-1RAs, which will lead to better risk-benefit evaluations in medical practice.

### 3.1. Adverse Dermatological Effects of GLP-1RAs

The administration of GLP-1RAs can lead to various skin reactions that range from injection-site localized responses to serious immune system-triggered conditions such as systemic hypersensitivity reactions (refer to Table 1).

#### 3.1.1. Injection-Site Reactions

GLP-1RAs most frequently cause dermatological adverse effects through injection-site reactions. Erythema and swelling along with nodular indurations and lipodystrophy represent the main injection-site reactions reported [46]. Lipodystrophy presents through localized fat loss or gain that can modify how drugs are absorbed and how effective they are [45]. Injection-site nodules that are generally harmless can become bothersome to patients and require stopping therapy in extreme cases [35]. Although most adverse reactions resolve without intervention some require treatment adjustments, which highlights the importance of correct injection methods and site variation [35].

Chronic inflammatory responses at the injection site have been reported due to prolonged subcutaneous GLP-1RA use [32]. Histological analyses demonstrate perivascular eosinophilic infiltration and granuloma formation, which indicate possible long-term tissue damage [46]. Individuals suffering from continuous injection-site inflammation should consider either changing their medication formulation or improving their injection methods.

#### 3.1.2. Systemic Hypersensitivity Reactions

Glucagon-like peptide-1 receptor agonists demonstrate growing acceptance as therapeutic agents for both glycemic regulation and weight control in patients with type 2 diabetes and obesity. As GLP-1 receptor agonists become more widely used, physicians can better identify adverse reactions, including systemic hypersensitivity responses. Medical professionals face significant challenges when managing patient reactions that display a spectrum of severity from mild rashes and localized urticaria to severe anaphylactic responses [2]. The extensive use of these medications requires clinicians to understand potential hypersensitivity reactions in order to teach patients how to identify early symptoms.

Severe dermatologic conditions such as generalized exanthematous pustulosis and acute photodistributed exanthematous pustulosis have been linked to the GLP-1RA liraglutide [47]. The clinical care of patients who have experienced drug allergies requires enhanced monitoring measures [51]. A recent medical case report described an eczematous skin reaction triggered by liraglutide controlled through dupilumab treatment, which targets interleukin-4 and interleukin-13 pathways. The case demonstrates the intricate management required for GLP-1RA-induced hypersensitivity reactions while showing the critical need for tailored treatment strategies [48].

Clinical data shows that Exenatide—a common GLP-1RA—has been associated with systemic allergic reactions, which emphasizes the need for meticulous patient selection and monitoring. The occurrence of systemic reactions and anaphylaxis indicates these infrequent incidents likely remain underreported during clinical treatments [50]. Post-marketing surveillance data analysis reveals that medical professionals frequently fail to identify or record anaphylactic reactions to GLP-1RAs immediately, resulting in underestimated incidence rates [4]. Healthcare providers must teach patients about identifying early hypersensitivity symptoms like pruritus, angioedema, and respiratory distress due to the severity of these reactions, which require quick medical intervention.

The scientific community has investigated potential connections between GLP-1RAs and autoimmune blistering diseases. A study found that GLP-1 analogs and sodium–glucose co-transporter-2 inhibitors do not raise bullous pemphigoid risk [57], while other studies indicate possible connections [42,43]. BP represents an autoimmune blistering condition that shows growing connections to GLP-1RA usage. Recent case reports indicate GLP-1RA therapy might trigger BP onset and cause drug-induced autoimmune reactions. The development of BP has been reported in patients using dulaglutide [42], semaglutide [43], and liraglutide [44], which suggests that various drugs within this class can induce BP. More studies are necessary to determine how this condition develops, and which factors make certain individuals more vulnerable. Medical professionals need to watch for BP in patients using GLP-1RAs when they develop new blistering skin lesions. Dulaglutide use has been linked to uncommon skin reactions beyond BP that include instances of pyoderma gangrenosum and antibiotic-resistant cellulitis at injection sites [33,34]. The research outcomes highlight the importance of continuous patient safety checks while GLP-1RAs are administered [58]. The range of effects among GLP-1 receptor agonists requires dermatological side effects evaluation through analysis of the literature (see Table 2).

The development of new GLP-1RA therapies requires post-marketing safety evaluations and real-world evidence to identify and address potential risks. Further research is needed to understand GLP-1RA-induced hypersensitivity mechanisms and to establish evidence-based guidelines for reaction management. Healthcare providers need to exercise heightened caution when prescribing these agents to patients who have experienced allergic reactions to medications in the past.

### 3.2. GLP-1RAs and Immune-Mediated Skin Conditions

The role of GLP-1RAs in immune-mediated skin disorders such as Pso and HS remains controversial. Some studies suggest a beneficial effect, while others report exacerbations [65] (see Table 3).

#### 3.2.1. Psoriasis and GLP-1RAs

Psoriasis represents a persistent immune-based skin condition that affects about 2–3% of people worldwide and shows strong links to systemic inflammation, metabolic dysfunction, and cardiometabolic health issues [67]. The connection between Pso with obesity and metabolic disorders has generated increased research interest in using GLP-1RA for Pso treatment.

Several investigations have explored how GLP-1RAs affect Psoriasis results. Median Psoriasis Area and Severity Index (PASI) scores fell from 21 to 10 across a 12-week period in a randomized clinical trial, which shows that semaglutide reduces Pso severity and improves Dermatology Life Quality Index (DLQI) results [58]. A prospective study demonstrated that 12 weeks of liraglutide treatment produced significant PASI score reductions from 15.7 to 2.2 and improved DLQI scores from 21.8 to 4.1 [60]. Clinical reports show that GLP-1RA therapy benefits Pso patients as evidenced by a 92% PASI reduction and full DLQI score normalization after 10 months of semaglutide treatment in a patient with severe Pso and obesity. Recent case study results show quick symptom improvement, with itching and scaling stopping shortly after starting liraglutide treatment, which points to anti-inflammatory qualities separate from weight loss benefits.

Basic science and clinical research evidence indicates that GLP-1RAs produce direct immunomodulatory actions that function separately from weight loss. This treatment results in lowered TNF-α [8] and IL-17 levels [8], blocks NF-κB signaling pathways [8,55], and decreases both C-reactive protein [19] and IL-6 levels [19]. Histological examinations show that treatment resulted in decreased epidermal thickness [60]. The observed clinical benefits in Pso patients treated with GLP-1RAs can be explained through these mechanistic findings (see Figure 4).

Conflicting evidence about Pso worsening with GLP-1RA treatment suggests differences in individual immune responses as a cause [65]. Some case reports show that GLP-1RA treatment initiation in patients leads to worsened Pso lesions, possibly because the drug triggers immune dysregulation, which results in increased pro-inflammatory pathways [19,65]. A meta-analysis shows that 35% of patients reached a 75% improvement in PASI scores following 6 to 12 weeks of GLP-1RA therapy, but many patients show minimal improvement demonstrating response variability [66]. Some case reports have revealed that Pso symptoms can deteriorate soon after patients start GLP-1RA therapy, which indicates that these medications may activate immune dysregulation instead of providing suppression in certain individuals [20]. The research results emphasize the necessity for customized treatment strategies and additional investigations to understand the causes of these adverse effects.

New research shows that GLP-1RAs may become beneficial additional treatments for psoriasis, especially in patients who suffer from metabolic conditions like obesity and T2DM. The long-term safety and effectiveness of GLP-1RAs for psoriasis needs confirmation through subsequent extensive clinical trials [60]. Together, dermatologists and endocrinologists can improve treatment approaches for patients with both metabolic conditions and inflammatory skin diseases [60,61]. Future studies into GLP-1RAs’ immunomodulatory effects will likely expand their dermatological uses.

#### 3.2.2. Hidradenitis Suppurativa and GLP-1RAs

The chronic skin disorder Hidradenitis suppurativa demonstrates a strong relationship with obesity. Weight loss approaches along with GLP-1RA therapy have been investigated as potential treatments due to metabolic dysfunction’s involvement in HS development. GLP-1RAs demonstrate anti-inflammatory and metabolic effects, making them a hopeful therapy choice for obese HS patients.

According to multiple reports, initial findings show that weight loss and immune system changes triggered by GLP-1RA treatment lead to fewer lesions and less frequent HS flares [63]. The systematic review of GLP-1RA use in HS patients demonstrated significant reductions in inflammatory nodules, abscess formation, and overall disease severity [62]. Despite the positive results observed so far, initiating GLP-1RA treatment may lead to increased inflammation in some individuals. A subset of patients who use GLP-1RAs experience HS flares, which points to a complex immunological reaction that differs based on individual immune profiles [62,65]. Research shows that GLP-1RAs have dual anti-inflammatory and pro-inflammatory effects in Psoriasis patients, which highlights the importance of customized treatment plans for each patient [58].

Apart from its effects on HS treatment, GLP-1RA therapy has been studied because of its connection with acne vulgaris (AV). The results of a recent meta-analysis show that GLP-1RA treatment leads to higher occurrences of AV, prompting concern about their comprehensive skin-related side effects [68]. Studies have demonstrated that both acne vulgaris (AV) and hidradenitis suppurativa (HS) share inflammatory pathways that result in immune system dysregulation through enhanced activity of IL-1β, TNF-α, and IL-17 pathways leading to follicular occlusion along with neutrophilic inflammation and abnormal keratinocyte proliferation [8]. The central role of these pathways in both conditions might explain the dual potential of GLP-1RAs to either worsen or improve disease progression based on the individual patient’s immune response and metabolic status. More research is required to determine if GLP-1RA treatment worsens or improves HS symptoms in patients who already have AV and HS.

### 3.3. Ozempic Face: Dermatological Impact of Rapid Weight Loss

The term “Ozempic face” describes how rapid weight loss through GLP-1 receptor agonists used for obesity and T2DM causes facial alterations. The condition reduces facial volume because of subcutaneous fat loss, creating a hollowed look with sunken features alongside deeper wrinkles and skin that sags. These changes mimic typical aging but happen suddenly in people who are still relatively young [6].

The precise mechanism behind “Ozempic face” remains unclear, but the quick weight loss from semaglutide treatment is thought to play a crucial role in these aesthetic changes. Research indicates that the medication promotes systemic fat reduction rather than directly affecting facial fat cells, which results in a gaunt facial appearance and signs of premature aging [55].

The rise of semaglutide usage correlates with increased dermatology and plastic surgery consultations where patients seek to restore facial volume after losing weight due to GLP-1RA treatment [52,53]. The increased need for facial volume restoration after GLP-1RA-induced weight loss has made corrective aesthetic procedures more popular, including dermal fillers, biostimulatory agents, skin-tightening treatments, and energy-based devices to address fat depletion effects [54].

Recent studies propose that collagen breakdown and skin elasticity alterations play a role in the facial hollowing frequently seen in patients using Ozempic [21]. Treatments that focus on collagen preservation, including retinoid skincare products, biostimulatory injectables, and radiofrequency microneedling, can provide additional advantages for sustaining skin health and postponing facial aging signs [52,55].

The expression “Ozempic face” lacks medical validation but highlights the cosmetic effects that arise from weight loss through GLP-1RA treatment. Healthcare providers must actively teach patients about possible side effects from GLP-1 agonists and create personalized multimodal treatment plans while discussing strategies to prevent unwanted changes as these medications become more commonly used for weight loss. When healthcare providers optimize treatment strategies that target both metabolic and esthetic outcomes, patients will better manage the side effects of GLP-1RA therapy and avoid unwanted facial volume loss.

## 4. Discussion

Increased use of GLP-1 receptor agonists for managing T2DM and obesity has heightened recognition of their skin-related side effects. These medications provide substantial advantages for weight loss and metabolic control but also produce multiple skin-related side effects. Clinical studies show increasing skin-related side effects associated with GLP-1RA treatment, such as hypersensitivity reactions and injection-site reactions along with immune-mediated conditions and facial fat loss [31,54]. Dermatological side effects from these medications can be mild and self-limiting or severe enough to necessitate stopping treatment [14]. Optimizing treatment strategies requires understanding these effects because GLP-1RAs are frequently prescribed off-label to aid in weight loss [6].

The requirement for GLP-1RAs to be administered through subcutaneous injection changed as advancements created new ways to deliver these medications. The approval of an oral semaglutide formulation in 2019 introduced the first GLP-1RA available for oral administration in clinical settings [67]. The oral semaglutide tablets integrate an absorption enhancer called SNAC, which safeguards the peptide against stomach conditions and promotes its absorption in the small intestine, thus achieving sufficient bioavailability [63]. Patients now have access to a GLP-1 therapy option that eliminates needles.

Research continues to advance transdermal delivery methods as alternatives to oral or subcutaneous medication options. Painless administration of large peptide drugs through the skin is possible with microneedle patches [69]. The HeroPatch system, among other available dissolvable microneedle patches, achieved successful weekly delivery of GLP-1RA doses in preclinical studies without injections. Delivery of GLP-1RA through transdermal methods avoids the gastrointestinal system, which may boost patient compliance if this technique proves effective in human applications [69]. Although investigational methods such as intranasal or inhaled formulations are still in their initial development phase, they illustrate a widespread initiative to develop alternatives to injection-based treatments. Although subcutaneous injection continues as the standard treatment approach, oral GLP-1RA therapy, together with new delivery technologies such as transdermal patches, demonstrate the arrival of more convenient administration options for this class of medication [67].

### 4.1. Adverse Associations with GLP-1RAs

The most common dermatological adverse effects of GLP-1RAs occur at the injection site and present as localized erythema along with swelling and nodular indurations [29]. Despite its rarity, lipodystrophy represents a serious medical issue because it can interfere with drug uptake while also causing aesthetic problems [45]. Exenatide has been linked to hypersensitivity reactions ranging from mild urticaria to life-threatening systemic allergic responses along with anaphylaxis [35,50]. Patients sometimes develop generalized exanthematous pustulosis when using liraglutide, which requires prompt recognition and discontinuation in severe cases.

The immunogenic potential of GLP-1RAs becomes apparent through morbilliform drug eruptions, especially with dulaglutide. According to a recent study, patients with the condition developed widespread maculopapular rashes, which disappeared when therapy was stopped. Biopsy results show perivascular lymphocytic infiltrates, which established the delayed hypersensitivity reaction mechanism [38]. Systemic corticosteroids became necessary for treating severe cases of injection-site reactions, which typically would resolve on their own. Early detection of adverse reactions prevents unnecessary treatment continuation and patient discomfort from extended exposure to therapy [41].

Recent meta-analytical findings indicate a possible connection between GLP-1RA therapy and acne vulgaris. The meta-analysis of AV occurrence in GLP-1 agonist-treated patients emphasized the requirement for additional research into the mechanisms behind this relationship. Researchers believe GLP-1RAs could affect sebaceous glands or change inflammatory responses that cause acne, though the exact physiological mechanisms remain undetermined [62,63,64]. Reports show that patients starting GLP-1RA treatments sometimes develop new acne or worsen existing acne conditions, which implies that these medications could contribute to acne development [68]. As GLP-1RAs become more commonly used, clinicians need to monitor for AV dermatological side effects while planning suitable treatment approaches for patients who develop this condition.

### 4.2. GLP-1RAs and Immune-Mediated Dermatological Conditions

Research indicates GLP-1RAs could play a dual function in immune-related skin disorders by showing both adverse side effects and possible therapeutic benefits. Concerns about dulaglutide’s immunomodulatory effects have emerged from isolated bullous pemphigoid and pyoderma gangrenosum cases [34,42]. Reports from separate cases have shown that dulaglutide can cause PG and BP, which adds to the expanding research on autoimmune skin conditions related to GLP-1RA. Retrospective studies alongside pharmacovigilance reports show no significant rise in autoimmune blistering diseases among users of GLP-1RA compared to other antidiabetic drugs, implying these reactions may be unique to specific patients [43,57]. We need additional research to understand these associations and identify patients predisposed to dermatologic autoimmune responses to GLP-1RA treatment.

New research demonstrates that GLP-1RAs possess immunomodulatory effects in autoimmune disorders. The GLP-1Rs expressed on immune cells such as macrophages, invariant natural killer T (iNKT) cells, and intestinal intraepithelial lymphocytes (IELs) modulate inflammatory pathways [70]. Exenatide elevated anti-inflammatory M2 macrophage polarization by 3.2 times (*p* < 0.01) while diminishing interferon-gamma and IL-4 levels in iNKT cells by 42% and 35%, respectively (*p* < 0.05). GLP-1R activation led to a reduction in IL-6 and TNF-α levels in IELs by as much as 50%, according to a study [70]. Research indicates that GLP-1RAs can provide additional therapeutic support for autoimmune skin conditions, including Pso and HS, through immune response regulation [61]. Researchers must perform additional clinical studies to confirm these effects and determine how to use them in therapy.

Scientists are investigating whether GLP-1 receptor agonists impact cellular stress responses within skin cells through the production of heat shock proteins (HSP) such as Hsp70. Hsp70 functions as a stress-induced chaperone acting as a “danger signal” when it is released outside cells and plays a role in chronic inflammatory skin conditions including psoriasis and lupus [71]. Current research fails to provide direct in vivo evidence demonstrating that GLP-1 receptor activation affects Hsp70 levels in human skin. Research outside of clinical settings shows that GLP-1 has the potential to control HSP levels in various bodily tissues. The GLP-1 analog exendin-4 demonstrated upregulation of Hsp72 (Hsp70) and activation of extracellular signal-regulated kinase (ERK) pathways in pancreatic beta-cell lines under stress conditions according to study [71]. GLP-1 analog treatment resulted in elevated Hsp70 levels in rodent neuronal tissue during injury states, which suggests activation of a protective heat-shock response [72]. Since keratinocytes serve as a primary inducible source of Hsp70 for skin immunity, we can imagine GLP-1RAs might influence these pathways indirectly through anti-inflammatory effects or the reduction of oxidative stress which normally lead to HSP production [71] (Figure 5). Existing research has not verified the induction of HSP70 in skin by GLP-1RA treatment; therefore, speculation continues regarding GLP-1’s involvement in skin repair or pathology through chaperone mechanisms. The potential effects of GLP-1RA on immunomodulation involving Hsp70 mechanisms in inflammatory skin diseases suggests an area for future research beyond cytokine interactions.

Researchers continue to explore the positive effects of GLP-1RAs on immune-driven skin diseases such as psoriasis. Research indicates that Pso has strong connections to metabolic syndrome and insulin resistance, while GLP-1RAs might provide anti-inflammatory benefits through the TNF-α and IL-17 pathways [8,62]. Research studies have shown that liraglutide and semaglutide treatment improves Pso patient outcomes and notable PASI score reductions [18,19]. A randomized controlled trial showed that patients undergoing liraglutide therapy exhibited reduced systemic inflammation and skin lesion severity, which supports its adjunctive use in Pso management [59]. The onset of GLP-1RA treatment can lead to Pso flare-ups in some patients, which indicates differences in how individuals’ immune systems respond to the medication [58,66]. Research through additional clinical trials should seek to discover which patient subgroups might benefit from GLP-1RAs as a treatment option for Pso.

GLP-1RAs show potential as a treatment for HS because they promote weight loss and exert anti-inflammatory effects. According to recent research, obesity-related HS patients experience less disease severity through weight reduction [63]. A recent systematic review indicated that HS patients on GLP-1RA treatment showed a reduction in inflammatory nodules and disease flares [62]. The research findings require validation through longitudinal studies, since GLP-1RAs might modify immune pathways in ways that could intensify inflammation for some people [51].

Researchers recently examined how GLP-1 affects invariant natural killer T cells and found that this medication might regulate immune equilibrium beyond its metabolic effects [68]. Clinical trials show that GLP-1RAs demonstrate therapeutic potential for Pso, especially when paired with traditional systemic treatments like acitretin [17,57]. We still do not understand how GLP-1RA therapy modulates psoriasis, but additional randomized controlled trials will help identify which patient groups can benefit from this treatment.

### 4.3. GLP-1RAs and Skin Cancer

The latest studies have investigated the possible link between GLP-1RAs and skin cancer development. Researchers conducted a population-based cohort study to determine if GLP-1RA use results in higher melanoma and nonmelanoma skin cancer risk than sulfonylureas in T2DM patients [73]. The research tracked 11,786 new GLP-1RA users and 208,519 new sulfonylurea users for melanoma evaluation and 11,774 new GLP-1RA users and 207,788 new sulfonylurea users for nonmelanoma skin cancer evaluation. The study showed that GLP-1RAs did not show increased melanoma risk (HR 0.96, 95% CI 0.53–1.75) or nonmelanoma skin cancer risk (HR 1.03, 95% CI 0.80–1.33) when compared to sulfonylureas. A link between cumulative duration of use and any skin cancer was not supported by any evidence [73].

Healthcare providers must focus on educating patients about GLP-1RA-related skin reactions and closely monitor them to achieve the best treatment results. Clinicians must review the histories of patients with skin problems, allergies, and autoimmune diseases before starting treatment to determine individuals with a greater likelihood of developing skin complications [9]. Medical professionals must educate patients about potential injection-site reactions and hypersensitivity reactions, which could lead to autoimmune conditions like pyoderma gangrenosum and BP [33,41,73]. Rapid weight loss patients need counseling on potential aesthetic issues and preventive actions such as skin hydration, collagen support, and facial strengthening exercises [27,51].

This review summarizes the dermatological effects linked to GLP-1RA treatment but leaves several questions unanswered. Future research must explore the underlying processes that cause skin alterations while identifying risk markers and finding ways to prevent adverse effects and maintain metabolic advantages. The future research agenda needs to include long-term studies to determine the permanence of facial fat loss effects and evaluate GLP-1RAs’ potential use against autoimmune skin disorders like psoriasis and bullous pemphigoid.

## 5. Conclusions

The systematic review demonstrates that GLP-1 receptor agonists cause various skin issues, from mild reactions like pruritus and rashes to serious conditions, including angioedema, eczema, and bullous pemphigoid. Several significant dermatological associations exist with conditions such as psoriasis and pyoderma gangrenosum, along with injection site reactions hidradenitis suppurativa and diffuse allergic reactions. Hair loss, acne vulgaris emergence, and possible skin cancer connections demand further scientific exploration. The majority of skin reactions from treatment resolve on their own, but some instances may require patients to stop treatment. Research shows that different GLP-1 agonists produce varying dermatological effects, which require individualized patient treatment strategies. Additional research is needed to understand skin reaction mechanisms and create standardized treatment protocols. Medical professionals must recognize these potential side effects to diagnose them promptly and intervene correctly, which helps achieve the best patient safety and treatment results.

## Figures and Tables

**Figure 1 diseases-13-00127-f001:**
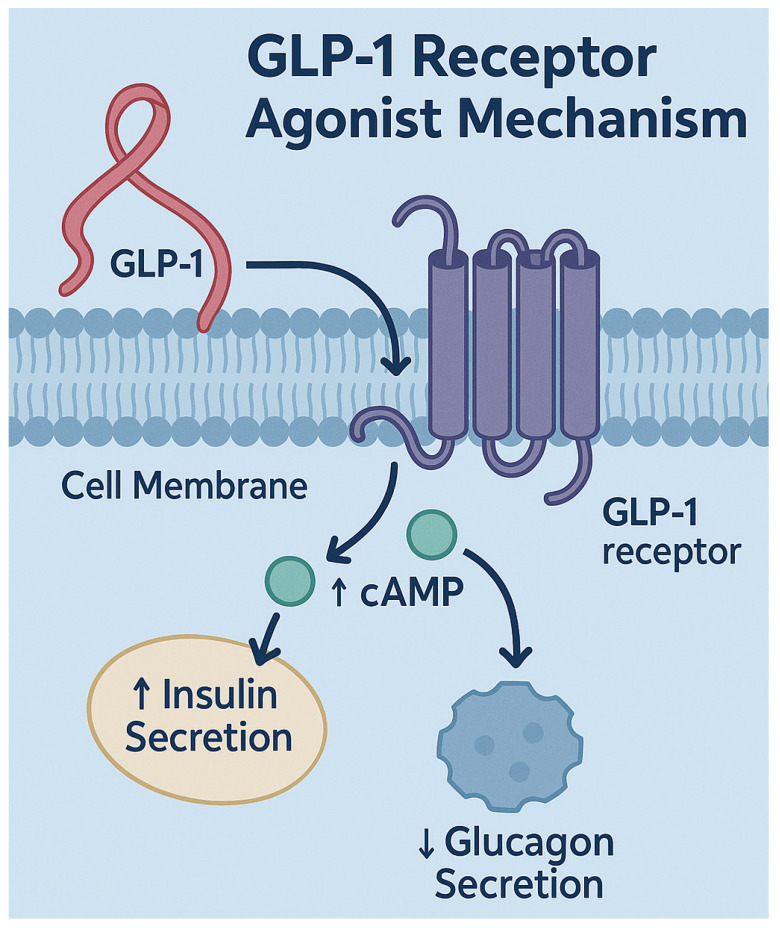
The figure illustrates the mechanism of GLP-1RAs. It shows GLP-1 binding to its receptor on pancreatic beta cells, activating intracellular signaling via adenylate cyclase and increasing cAMP. This triggers pathways involving PKA and Epac2, leading to enhanced insulin secretion, reduced glucagon release, delayed gastric emptying, and appetite suppression. The effects are depicted with arrows and labels across pancreatic, gastrointestinal, and central nervous system targets.

**Figure 2 diseases-13-00127-f002:**
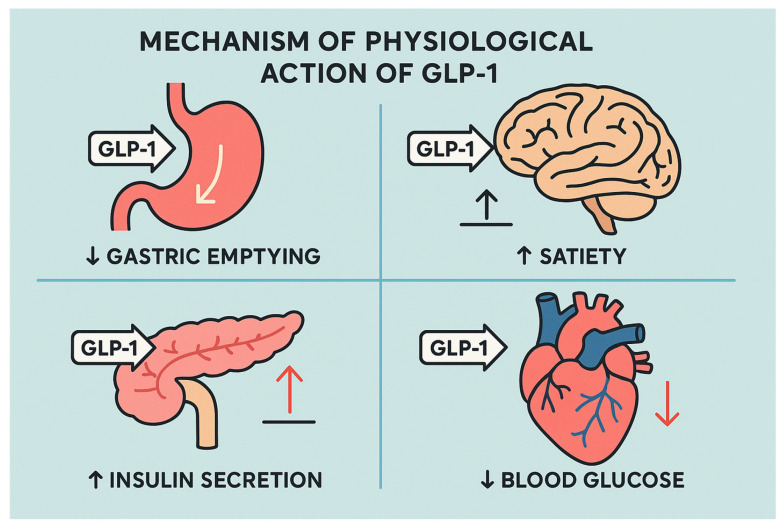
The figure illustrates the physiological actions of GLP-1 in a clear, flat-design style. It shows that GLP-1 stimulates insulin secretion from the pancreas, delays gastric emptying, and promotes satiety by acting on the brain. It also helps lower blood glucose by reducing hepatic glucose output. Together, these actions support appetite control and improved glycemic regulation.

**Figure 3 diseases-13-00127-f003:**
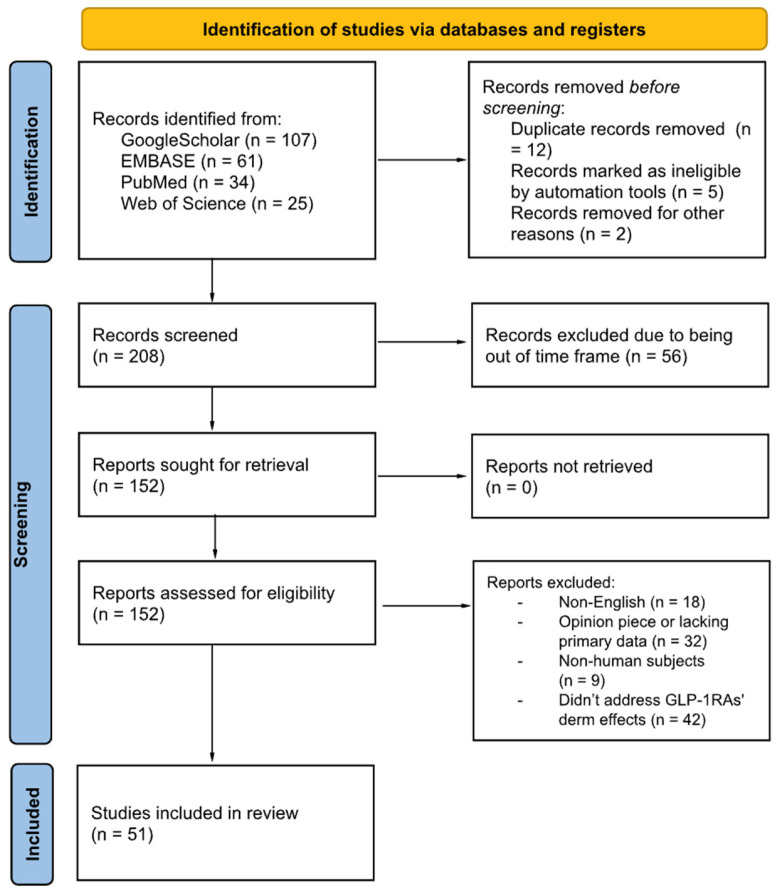
PRISMA flowchart.

**Figure 4 diseases-13-00127-f004:**
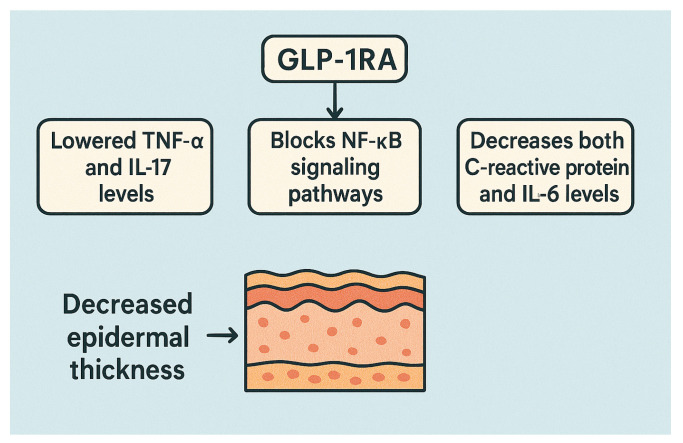
Immunomodulatory effects of GLP-1RAs. This figure illustrates the immunomodulatory effects of GLP-1 receptor agonists, independent of weight loss. It shows that GLP-1RAs reduce pro-inflammatory markers such as TNF-α, IL-17, IL-6, and C-reactive protein, while also blocking the NF-κB signaling pathway. A histological skin section highlights decreased epidermal thickness, linking these molecular effects to observed clinical improvements in conditions like psoriasis.

**Figure 5 diseases-13-00127-f005:**
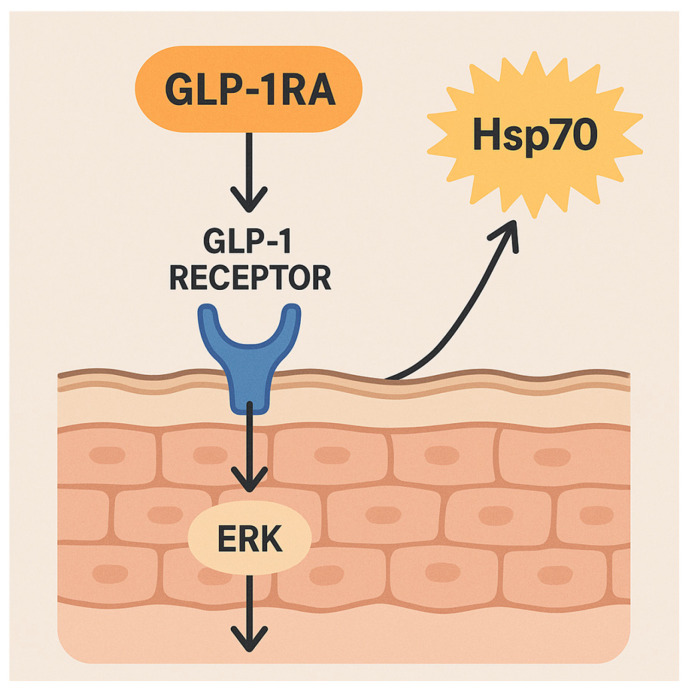
Potential GLP-1RA influence on Hsp70 in skin cells. The figure shows a cross-section of skin highlighting keratinocytes and the proposed effect of GLP-1 receptor agonists on Hsp70 expression. GLP-1RAs bind to the GLP-1 receptor, activating the ERK signaling pathway, which may lead to increased Hsp70 production. Hsp70, shown both inside and outside cells, functions in cellular protection and immune signaling. While direct skin evidence is lacking, the diagram illustrates a speculative pathway linking GLP-1RA, ERK activation, and heat shock protein-mediated immune modulation.

**Table 1 diseases-13-00127-t001:** Summary of adverse dermatological effects of GLP-1RAs.

Adverse Effect	Number of Reported Cases ^1^	References
Injection-site reactions	35	[29,31,32,33,34,35,36,37]
Pruritus	28	[1,10,14,15,31,36]
Rash	22	[1,2,10,14,29,31]
Angioedema	10	[2,14,17,20]
Urticaria	15	[1,2,14,17]
Hypersensitivity reactions	18	[14,17,31,37,38,39,40]
Morbilliform drug eruption	2	[38,41]
Bullous pemphigoid	3	[42,43,44]
Pyoderma gangrenosum	1	[34]
Lipodystrophy	12	[29,32,35,37,45,46]
Generalized exanthematous pustulosis	6	[36,47,48,49,50]
Anaphylaxis	5	[14,17,20]
“Ozempic Face”	9	[21,22,51,52,53,54,55,56]

^1^ These numbers only include reported cases from selected studies and may not reflect the event’s overall frequency due to varying study designs and reporting practices.

**Table 2 diseases-13-00127-t002:** Comparison of reported dermatological side effects of different GLP-1Ras.

GLP-1RA	Injection-Site Reactions ^1^	Hypersensitivity Reactions ^1^	Psoriasis Improvement ^1^	Hidradenitis Suppurativa Improvement ^1^	References
Exenatide	12	5	2	1	[14,35,46,50]
Liraglutide	8	6	4	3	[18,27,31,32,36,37,39,47,48,59,60,61,62,63,64]
Dulaglutide	10	5	1	2	[17,20,33,34,38,40,41,42,64]
Semaglutide	5	4	3	2	[19,20,25,29,43,58,60,61,62,63,64]

^1^ These numbers only include reported cases from selected studies and may not reflect the event’s overall frequency due to varying study designs and reporting practices.

**Table 3 diseases-13-00127-t003:** Summary of potential dermatological benefits of GLP-1RAs.

Beneficial Effect	Number of Supporting Cases ^1^	References
Psoriasis improvement	10	[18,19,59,62,66]
Reduced hidradenitis suppurativa flares	7	[24,62,63,64]
Enhanced wound healing	4	[18,24,27]
Potential anti-aging properties	3	[6,11,13]
Reduced skin inflammation	6	[11,18,19,59,62]

^1^ These numbers only include reported cases from selected studies and may not reflect the event’s overall frequency due to varying study designs and reporting practices.

## Data Availability

No new data were created or analyzed in this study. Data sharing is not applicable to this article.

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
