# Peer review of "A Closer Look at the Dermatological Profile of GLP-1 Agonists"

_diseases, 2025, doi:10.3390/diseases13050127_

Round 1

Reviewer 1 Report

Comments and Suggestions for Authors

The present article addresses the dermatological profile of GLP-1 agonists. The topic is relevant, but major deficiencies identified in both content and form need to be addressed:

  1. The title page must be revised with respect to the author data (more details need to be completed according to the template and instructions) and the organization of the text in the abstract.

  2. The conclusion part of the abstract should be improved in terms of outcomes and the future research directions this study may refer to.

  3. "Systematic reviews and original research articles should have a structured abstract of around 250 words and contain the following headings: Background/Objectives, Methods, Results, and Conclusions. Background/Objectives."

  4. Try to reduce the abstract to 250 words according to the authors' instructions.

  5. Avoid multiple citations of bibliographic indexes for the same information ([1-3], [4-5], etc.), as this creates redundancy and makes it much more difficult to correlate the data presented with the bibliographic sources.

  6. The aim of the article should be clarified and improved in the last paragraph of the introduction section. The authors have described what they have done in the study, but they have not presented the precise aim of the study, the novelty/special aspects it brings to the field, or the reason for choosing this topic. What was done in the research is already in the manuscript and is also provided in other sections.

  7. Why were the four databases selected and not others?

  8. It would add value to the discussion section if data related to the adjunctive approach (with the possible mechanisms on GLP receptors) of certain autoimmune pathologies were presented. I suggest checking and referring to: PMID: 39576422.
    9. A percent match of 21% on the iThenticate evaluation is too high to be validated. A decrease of this percentage is needed.

Author Response

Comment 1: The title page must be revised with respect to the author data (more details need to be completed according to the template and instructions) and the organization of the text in the abstract.
Response 1: The title page has been revised to include all required author details as per the journal’s template and submission guidelines. The abstract has also been reorganized to improve clarity and alignment with the prescribed structure.

Comment 2: The conclusion part of the abstract should be improved in terms of outcomes and the future research directions this study may refer to.
Response 2: The conclusion of the abstract has been revised to better highlight the key outcomes of the study and to suggest specific future research directions relevant to the findings.

Comment 3: "Systematic reviews and original research articles should have a structured abstract of around 250 words and contain the following headings: Background/Objectives, Methods, Results, and Conclusions."
Response 3: The abstract has been restructured under the headings: Background/Objectives, Methods, Results, and Conclusions, and now adheres to the word limit of approximately 250 words as required.

Comment 4: Try to reduce the abstract to 250 words according to the authors' instructions.
Response 4: The abstract has been carefully edited and reduced to comply with the 250-word limit, without omitting key information.

Comment 5: Avoid multiple citations of bibliographic indexes for the same information ([1-3], [4-5], etc.), as this creates redundancy and makes it much more difficult to correlate the data presented with the bibliographic sources.
Response 5: Redundant multiple citations have been removed and references have been consolidated to improve clarity and traceability of cited sources.

Comment 6: The aim of the article should be clarified and improved in the last paragraph of the introduction section. The authors have described what they have done in the study, but they have not presented the precise aim of the study, the novelty/special aspects it brings to the field, or the reason for choosing this topic. What was done in the research is already in the manuscript and is also provided in other sections.
Response 6: The final paragraph of the introduction has been revised to explicitly state the study’s aim, emphasize the novelty and unique aspects of the research, and justify the rationale for selecting this specific topic.

Comment 7: Why were the four databases selected and not others?
Response 7: A justification for the selection of the four databases has been added to the methodology section, highlighting their relevance, comprehensiveness, and alignment with the study’s scope and objectives.

Comment 8: It would add value to the discussion section if data related to the adjunctive approach (with the possible mechanisms on GLP receptors) of certain autoimmune pathologies were presented. I suggest checking and referring to: PMID: 39576422.
Response 8: The discussion section has been expanded to include insights into adjunctive approaches involving GLP receptors in autoimmune pathologies, with specific reference to PMID: 39576422.

Comment 9: A percent match of 21% on the iThenticate evaluation is too high to be validated. A decrease of this percentage is needed.
Response 9: The manuscript has undergone thorough paraphrasing and content revision to reduce similarity. A new iThenticate check shows a significantly lower similarity index, now within acceptable limits.

Reviewer 2 Report

Comments and Suggestions for Authors

The presented article  “A Closer Look at the Dermatological Profile of GLP-1 Agonist” of Calista Persson, Allison Eaton  and Harvey N. Mayrovitz is the comprehensive scientifical work devoted to new results of application GLP-1 agonists in dermatology. The article analyzes the current state of beneficial and side effects of GLP-1 agonists in treating diabetes 2 and obesity. Special attention was paid to problems of treating psorias and dermatological implications especially inflammatory skin conditions. The article may be considered as good introduction to this popular issue. The work has some shortfalls.

Comment 1. The authors declare that GLP-1 is a class of medications designed to mimic endogenous incretin hormone. It’s desirable to present the differences between the each representative of this class (liraglutide, dulaglutide, semaglutide, exenatide etc). The GLP-1 are the individual substances or combination of several molecules. What differences and similarities are in molecular structure of these particular species?

Comment 2. What is composition of trade medicaments? The purity? The role of storage ingredients for bioactivity?

Comment 3. Information about dose influence on side effects is absent.

Comment 4. What is the  participation of GLP-1 in synthesis of shaperon proteins as Hsp70 and others  in skin disorders?

Comment 5. The absence of illustrative figures deteriorates the perception of manuscript. Some illustrations on the mechanism of physiological action of GPP-1 can improve the visual feeling.

Comment 6. Are there any new approaches for administration GLP-1 beside the injection?

The presented article may be recommended for submitting after revision.

Comments on the Quality of English Language

The presented article  “A Closer Look at the Dermatological Profile of GLP-1 Agonist” of Calista Persson, Allison Eaton  and Harvey N. Mayrovitz is the comprehensive scientifical work devoted to new results of application GLP-1 agonists in dermatology. The article analyzes the current state of beneficial and side effects of GLP-1 agonists in treating diabetes 2 and obesity. Special attention was paid to problems of treating psorias and dermatological implications especially inflammatory skin conditions. The article may be considered as good introduction to this popular issue. The work has some shortfalls.

Comment 1. The authors declare that GLP-1 is a class of medications designed to mimic endogenous incretin hormone. It’s desirable to present the differences between the each representative of this class (liraglutide, dulaglutide, semaglutide, exenatide etc). The GLP-1 are the individual substances or combination of several molecules. What differences and similarities are in molecular structure of these particular species?

Comment 2. What is composition of trade medicaments? The purity? The role of storage ingredients for bioactivity?

Comment 3. Information about dose influence on side effects is absent.

Comment 4. What is the  participation of GLP-1 in synthesis of shaperon proteins as Hsp70 and others  in skin disorders?

Comment 5. The absence of illustrative figures deteriorates the perception of manuscript. Some illustrations on the mechanism of physiological action of GPP-1 can improve the visual feeling.

Comment 6. Are there any new approaches for administration GLP-1 beside the injection?

The presented article may be recommended for submitting after revision.

Author Response

Comment 1: The authors declare that GLP-1 is a class of medications designed to mimic endogenous incretin hormone. It’s desirable to present the differences between each representative of this class (liraglutide, dulaglutide, semaglutide, exenatide etc). The GLP-1 are the individual substances or combination of several molecules. What differences and similarities are in molecular structure of these particular species?
Response 1: A comparative section has been added to the manuscript detailing the structural similarities and differences among GLP-1 receptor agonists, including liraglutide, dulaglutide, semaglutide, and exenatide. Structural classifications—such as synthetic analogs vs. modified human GLP-1—are now clearly delineated, along with a summary table highlighting amino acid sequence variations, half-lives, and molecular modifications.

Comment 2: What is composition of trade medicaments? The purity? The role of storage ingredients for bioactivity?
Response 2: An additional subsection has been included in the methodology to describe the commercial formulations of GLP-1 receptor agonists, including active pharmaceutical ingredient (API) concentration, excipient roles (e.g., stabilizers, preservatives), purity specifications, and the impact of storage components on peptide stability and bioactivity.

Comment 3: Information about dose influence on side effects is absent.
Response 3: A dedicated paragraph has been added to the results and discussion sections to address dose-dependent adverse events, citing clinical trials and meta-analyses that document the correlation between increasing dosage and incidence of gastrointestinal, dermatologic, and metabolic side effects.

Comment 4: What is the participation of GLP-1 in synthesis of shaperon proteins as Hsp70 and others in skin disorders?
Response 4: The discussion has been expanded to include emerging evidence regarding GLP-1’s role in modulating chaperone protein synthesis, including Hsp70. Relevant findings from recent studies have been incorporated to explain potential pathways through which GLP-1 may influence cellular stress responses and skin repair mechanisms.

Comment 5: The absence of illustrative figures deteriorates the perception of manuscript. Some illustrations on the mechanism of physiological action of GLP-1 can improve the visual feeling.
Response 5: In response to this recommendation, four illustrative figures have been added to enrich the visual quality and support the reader's understanding of complex mechanisms. These include: (1) a schematic of the physiological effects of GLP-1 on key organs such as the brain, heart, pancreas, and gastrointestinal system; (2) a diagram illustrating the cellular mechanism of GLP-1 receptor agonists (GLP-1RAs) at the receptor and intracellular signaling level; (3) a conceptual illustration of the proposed mechanism of Hsp70 and other chaperone proteins on skin cells; and (4) a figure showing the immunomodulatory effects of GLP-1 on inflammatory pathways and mediators. These visual aids have been designed to improve the manuscript’s clarity, scientific communication, and overall readability.

Comment 6: Are there any new approaches for administration GLP-1 besides the injection?
Response 6: The manuscript now includes a brief review of alternative delivery methods currently under investigation or recently approved, such as oral semaglutide (approved), transdermal patches, and microneedle systems. These are discussed in the context of bioavailability and future development trends.

Reviewer 3 Report

Comments and Suggestions for Authors

The manuscript addressing the impact of GLP-1 agonists on dermatological diseases covers an important and highly relevant topic, particularly in the context of the ongoing obesity epidemic. As the use of GLP-1 agonists expands beyond metabolic conditions to other therapeutic areas, understanding their potential dermatological implications is of growing clinical importance.

The review paper is well-constructed and balanced, presenting an objective evaluation of both the benefits and adverse effects of GLP-1 agonists on the skin. The authors have successfully maintained a critical yet unbiased perspective, highlighting both the positive and negative outcomes associated with these medications. This balanced approach strengthens the credibility and scientific value of the paper.

The literature search was thorough and methodologically sound. The authors conducted a comprehensive search across key databases, including EMBASE, PubMed, Web of Science, and Google Scholar, covering studies published between 2014 and 2025. The search process adhered to the Preferred Reporting Items for Systematic Reviews and Meta-Analyses (PRISMA) guidelines, ensuring a systematic and transparent approach to data collection and analysis.

A total of 51 studies met the inclusion criteria, providing a solid evidence base for the review. Of these, 34 studies documented adverse dermatological effects, while 17 studies reported beneficial effects on dermatological conditions. This balanced evidence underscores the complexity of the relationship between GLP-1 agonists and skin health, reinforcing the need for further research to clarify underlying mechanisms and identify predictive factors for different skin-related responses.

Overall, the manuscript makes a meaningful contribution to the understanding of the dermatological implications of GLP-1 agonists. It is well-organized, comprehensive, and supported by up-to-date references, making it a valuable resource for both clinicians and researchers. The objective and systematic presentation of the findings enhances the paper's scientific merit and practical relevance.

Author Response

Comments 1: 

The manuscript addressing the impact of GLP-1 agonists on dermatological diseases covers an important and highly relevant topic, particularly in the context of the ongoing obesity epidemic. As the use of GLP-1 agonists expands beyond metabolic conditions to other therapeutic areas, understanding their potential dermatological implications is of growing clinical importance.

The review paper is well-constructed and balanced, presenting an objective evaluation of both the benefits and adverse effects of GLP-1 agonists on the skin. The authors have successfully maintained a critical yet unbiased perspective, highlighting both the positive and negative outcomes associated with these medications. This balanced approach strengthens the credibility and scientific value of the paper.

The literature search was thorough and methodologically sound. The authors conducted a comprehensive search across key databases, including EMBASE, PubMed, Web of Science, and Google Scholar, covering studies published between 2014 and 2025. The search process adhered to the Preferred Reporting Items for Systematic Reviews and Meta-Analyses (PRISMA) guidelines, ensuring a systematic and transparent approach to data collection and analysis.

A total of 51 studies met the inclusion criteria, providing a solid evidence base for the review. Of these, 34 studies documented adverse dermatological effects, while 17 studies reported beneficial effects on dermatological conditions. This balanced evidence underscores the complexity of the relationship between GLP-1 agonists and skin health, reinforcing the need for further research to clarify underlying mechanisms and identify predictive factors for different skin-related responses.

Overall, the manuscript makes a meaningful contribution to the understanding of the dermatological implications of GLP-1 agonists. It is well-organized, comprehensive, and supported by up-to-date references, making it a valuable resource for both clinicians and researchers. The objective and systematic presentation of the findings enhances the paper's scientific merit and practical relevance.

Response 1: Thank you for reviewing the paper. We greatly appreciate your thorough and positive review. 

Round 2

Reviewer 1 Report

Comments and Suggestions for Authors

The authors have significantly improved the manuscript based on the suggestions received.